# Position: Want Better ML Reviews?
# Stop Asking Nicely and Start Incentivizing with a Credit System

**Shaochen (Henry) Zhong** [1]

## Abstract

With soaring submission counts, stricter reciprocal review policies, widespread adoption of platforms like OpenReview, and without the offsetting pressure of publication fees, the machine learning (ML) community has one of the largest scholarly presences among all scientific fields. And yet, **almost *everyone* has *many* unpleasant things to share about their review experience.** Worse, there is little public space to seriously discuss, let alone debate, what makes a review system effective or how it might be improved. In this position paper, we expand our discussion from two core problems: *How can we reasonably limit the number of submissions?* and *How can we incentivize good and discourage bad review practices?* We first assess the strengths and shortcomings of existing attempts to address such problems. Specifically, we present four takes on some popular conference mechanisms and propose two alternative designs for improvement. Our general position is that meaningful improvement in ML peer review won't come from polite best-practice suggestions tucked into Calls for Papers or Reviewer Guidelines: it requires **enforceable yet fine-grained procedural safeguards** paired with **a currency-like credit system (e.g., our proposed *OpenReview Points*)**. ML practitioners can "earn" such points by contributing good review practices, and "spend" across one or multiple major conferences to redeem different kinds of "perks," such as complimentary registration or the right to request additional review resources.

## 1. Introduction

**This position paper argues that peer review in machine learning (ML) is unlikely to improve through polite requests or optimistic guidance tucked into Calls for Papers or Reviewer Guidelines. Fine-grained yet enforceable procedural guardrails, combined with a spendable, across-conference credit system, are almost mandatory for a sustainable review ecosystem.**

Machine learning has scaled faster than nearly any other scientific field in both volume and visibility. We now have tens of thousands of paper submissions to a single conference (the most recent NeurIPS 2025 had 21,575 valid submissions to the main conference alone), open-access platforms like OpenReview that support interactive discussions, and increasingly reciprocal reviewing obligations to match supply with demand. On paper, the ML community has everything it needs to sustain a robust yet pleasant peer-review pipeline: we have the largest scholarly presence of any scientific field and the most modern review technology, all without the typical bottlenecks of paywalls, publication fees, or expensive memberships. However, the lived reality often feels far less functional. From cryptic or dismissive reviews to wildly inconsistent standards, frustrations with the review process appear widely shared (Thorn Jakobsen & Rogers, 2022), voiced by PhD students, seasoned professors, and industry researchers alike (e.g., see examples in Section 2.1).[1] Worse, there is little to no structured way to hold bad actors accountable, nor are there incentives to encourage good actors to go the extra mile.

In this position paper, we expand our discussion of the two core challenges we have identified:

1. ***How can we reasonably limit the number of submissions?***
2. ***How can we incentivize good and discourage bad review practices?***

We first lay the background on why these two issues are the root causes of much unpleasantness in ML review. Then,

---

[1]Department of Computer Science, Rice University, Houston, TX, USA. Correspondence to: Shaochen (Henry) Zhong <henry.zhong@rice.edu>.

*Proceedings of the 43rd International Conference on Machine Learning*, Seoul, South Korea. PMLR 306, 2026. Copyright 2026 by the author(s).

---

[1]In the NeurIPS 2025 Datasets & Benchmarks author survey, for instance, roughly 25% of respondents flagged review quality as needing improvement.

we assess some existing attempts to mitigate such issues as implemented in several ML conferences. We present our takes on such measures and, finally, propose two new mechanisms: **fine-grained procedural safeguards that could be enforced at scale; and a credit system based on something we call *"OpenReview Points"*, which would let researchers "earn" and "spend" their reviewing efforts in tangible ways across all major conferences and review cycles.** We believe such mechanisms would have a fair chance of addressing many of the aforementioned shortcomings effectively and, more importantly, are flexible enough to allow each conference to adopt its own variants. Outside our proposed mechanism, we engage many alternative views of ours, where we discuss how such views are valid (or not) and how our proposed mechanism shall be able to take such concerns into consideration. We conclude our paper with a *Recommended Practices* section, which outlines our vision on how the first few conferences adopting a similar credit system should proceed and what aspects should be considered cautiously. Additional materials, such as Frequently Asked Questions and anecdotal case studies based upon real conferences, can be found in Appendix B and Appendix C respectively. Due to limited immediately relevant work and page limits, we present related works in Appendix A.

We faithfully emphasize that our goal is not to perfect ML peer review (as it would be unfaithful and condescending for anyone to claim so), but to make its failures rarer, less painful, and, most importantly, more accountable and sustainable. We are also not there to propose a specific rulebook that all conferences must follow, but rather to advocate for a promising general direction that future conference organizers can explore and adapt to their own needs.

## 2. Root Causes

### 2.1. Overblown number of submissions causes all kinds of challenges.

We believe it is common knowledge that ML conferences typically receive an overblown amount of submissions (Kim et al., 2025; Yang, 2025). Naturally, this causes all kinds of practical challenges. From a manpower perspective, more submissions directly mean greater demand for reviewers and Area Chairs (ACs), which translates to a heavier workload for Senior Area Chairs (SACs) and, eventually, Program Chairs (PCs). With such great pressure on every aspect of the conference review system, the results are almost predictable: thinner attention per paper, more rushed triage, and greater variance in both review quality and decision outcomes (Su et al., 2025; Shah, 2022).

Moreover, practicality-wise, most ML conferences often guarantee the right to in-person presentation exposure once

a paper is accepted.[2] This makes physical capacity restrictions come into play, directly imposing an upper limit on how many papers can be accepted. Many borderline or acceptance-inclined papers might be ruled out purely based on capacity constraints, a role often delegated to SACs. But considering the number of submissions versus the number of SACs (per the NeurIPS 2024 fact sheet, 195 main-conference SACs against 15,671 submissions, roughly 80 papers per SAC), this kind of assignment is, by design, unreasonable and unsustainable, as few SACs would have the bandwidth or appetite to go through the content and review record of that many papers. In practice, this pressure incentivizes shortcutting (e.g., relying more heavily on numerical scores or other similar quick heuristics), which further amplifies randomness and weakens accountability. In fact, we have seen many SACs publicly pushing against such "force rejection for capacity" practices, as exemplified by LinkedIn posts from NeurIPS SACs Ahmad Beirami and Atlas Wang.

### 2.2. Lack of oversight, feedback loop, and incentives for good actors and consequences for bad ones.

While reviewers, ACs, and SACs have the right to provide feedback, ML conferences lack proper oversight and feedback loops. A reviewer can engage in highly discretionary actions so long as they are not extreme enough to trigger formal intervention. **This ecosystem leaves actors with few avenues to learn how to become better, let alone much incentive to go the extra mile.** Without routinized feedback, transparent metrics, or positive incentives, the system neither rewards exemplary stewardship nor deters poor practices; we elaborate on this in Section 3.4.

## 3. Why Existing Fixes Fall Short

### 3.1. Soft and hard submission caps offer limited help.

With the growing body of research in the ML community (Yang, 2025; Kim et al., 2025) and with AI-assisted research becoming more accessible[3] (Eger et al., 2025), the volume of submissions continues to grow at a pace that far outstrips the community's reviewing capacity. This escalation naturally prompts discussions around mechanisms for curbing submission rates and maintaining a manageable reviewing load, where submission caps are often proposed as one of

---

[2]That said, this tradition might be undergoing a serious update, with ICML 2026 no longer requiring in-person attendance.

[3]This is particularly evidenced by recent autoresearch tooling such as Karpathy's `autoresearch`, and by coding-assistant plans such as Claude Code moving Opus's 1M-token context from extra usage into the standard subscription, both lowering the barrier to AI-assisted work. We can even find entirely AI-driven work accepted at major NLP venues, e.g., an AI scientist getting a paper into the main track of ACL 2025 (see this blog and Zhou & Arel, 2025).

*Table 1.* **Most-submitted authors at ICLR 2025, counted by author position** per PaperCopilot statistics (Yang, 2025). Each cell reports the submission count of the $n$-th most prolific author under that counting. A handful of authors appear on many papers overall (*any author*), yet very few are *first* or *last* authors on comparably many, suggesting that for many heavily-listed authors, core involvement across all those papers is unlikely.

| Most submitted | #1 | #25 | #50 | #75 | #100 |
| --- | --- | --- | --- | --- | --- |
| First author | 7 | 4 | 3 | 3 | 3 |
| Last author | 34 | 13 | 11 | 9 | 8 |
| Any author | 42 | 21 | 17 | 16 | 14 |

the most direct ways to reduce such volume.

We argue that submission caps, whether "soft" (e.g., mandatory reciprocal review over X submissions) or "hard" (e.g., strict per-author quotas on how many papers can be submitted), provide, at best, marginal relief. The core problem is not that a small set of "hyper-prolific" lead authors are personally flooding the system with many new submissions (Yang, 2025), but that there is essentially no downside to submitting unready manuscripts or endlessly recycling previously rejected work with critical flaws. We suspect that, in practice, per-author caps mostly trim auxiliary authors from the byline so that teams can fit under the quota; they do little to stop the same lead authors from submitting the same number of papers, worthy or not, or from repeatedly resubmitting flawed work.

In other words, without penalties for low-quality submissions, **submission caps may largely change who gets listed on a paper, rather than whether the paper is submitted.** While we lack direct evidence, public statistics are at least suggestive: as shown in Table 1, many authors appear on a large number of ICLR 2025 submissions, yet very few are first or last authors on comparably many, making it unlikely that they are core contributors to all of them. Under a harsh cap, such authors might simply drop off lower-priority bylines rather than forgo a submission. Thus, we argue that until there is a genuine negative incentive that discourages unlimited resubmission and/or rewards restraint, submission caps can only nibble at the edges of the volume problem, instead of addressing its core.

### 3.2. Irresponsible Reviewers Care Most About Their Own Works, So Asking Nicely Might Not Be Helpful

The seemingly widespread presence of bad or irresponsible review practices stems in large part from the lack of accountability built into current conference mechanisms. Until very recently, most ML conferences enforced no retaliatory punishment against irresponsible reviewers, leaving bad practices essentially unchecked. For a reviewer who is recruited by force and treats the assigned duty as a mere

task to be finished, the things that matter most are likely their own current or future submissions. We argue that, to effectively discourage irresponsible reviewing, **some form of retaliatory penalty must be enforced at the submission end.** Otherwise, conferences have little real leverage and often can only resort to asking nicely in Calls for Papers or Reviewer Guidelines; and despite the existence of some extremely thoughtful guidelines like the ARR Reviewer Guideline, their effectiveness leaves much to be desired.

Recently, starting with CVPR 2025, several ML conferences have adopted what is essentially a retaliatory desk-rejection policy targeting irresponsible reviewers. At CVPR 2025, its Area Chairs (ACs) *"identified a number of highly irresponsible reviewers, those who either abandoned the review process entirely or submitted egregiously low-quality reviews, including some generated by large language models"* and ultimately issued desk rejections for 19 otherwise accepted papers involving those reviewers.

While this act marks a meaningful start to enforcing hard procedural guardrails to protect review quality and integrity, we argue that **retaliatory procedures as harsh as desk rejection can only offer marginal benefits to the conference at large, as only a few bad actors would be extreme enough to blatantly ignore direct instructions.** This is because a harsh penalty like desk rejection is, by nature, a blunt instrument: it is too harsh to apply broadly and can only be reasonably used for the most extreme violations with verifiable signals. In CVPR's case, it was mostly reserved for reviewers who outright abandoned their review duties, a clean, rule-based verifiable breach that leaves no room for ambiguity.

Unfortunately, most reviewer problems in ML are arguably more subtle than complete negligence. Irresponsible reviews can manifest in many forms: from thoughtless boilerplate complaints like "no theory" or "needs more experiments" applied indiscriminately to every submission, to a gross misunderstanding of basic facts and refusal to reconsider, to a raised concern with no concrete support, or even the famous "Who is Adam?". These reviews are much harder to police, but no less damaging.

We argue that desk rejection is too coarse a penalty to handle the long tail of poor reviewing behaviors that fall short of full abandonment. If we want meaningful deterrents at scale, we need a system that applies graduated, proportional penalties, not just all-or-nothing rulings. The fact that the CVPR 2025 procedure only resulted in 19 desk rejections suggests that the irresponsible review issue in the ML community is far from resolved by adopting this retaliatory desk rejection policy alone; more enforceable yet fine-grained procedural safeguards are necessary to handle the wide spectrum of irresponsible review practices. In other words, **desk rejections are like felony charges, but we also need parking**

**tickets and everything in between to moderate a proper review community.**

### 3.3. 100% Mandatory Reciprocal Reviewer Recruit is a Slow-Acting Poison

To keep up with rising submission counts, many ML conferences, starting with the most recent EMNLP 2025 (ARR May), now rely on 100% reciprocal reviewer recruitment: every eligible author[4] must also review, except in extreme circumstances like parental leave. On paper, this sounds fair: if one wants to publish, one should contribute to the review pool. But in practice, we argue it is a slow-acting poison that comes at the cost of review quality.

The policy assumes that all eligible authors of every accepted paper are both capable and willing to provide thoughtful reviews. That assumption does not hold across the ML community: many eligible authors may have played only auxiliary roles or contributed as expert consultants on highly specialized components. They are therefore ill-suited to reviewing general-purpose ML submissions. Forcing mandatory review duties on such contributors often produces a predictable outcome: low-effort reviews written mainly to avoid desk rejections. Worse, mandatory review removes the ability for people to decline, even when they know they cannot meaningfully contribute due to sensible (but non-medical) emergencies and circumstances. Once in the reviewer pool, conferences often allow very limited flexibility for exemption. For instance, AAAI 2026 instructed their reviewers to "do your best" even if the assigned paper is outside their area of expertise.

We argue that such enforced cultures would likely result in a series of rushed, templatized, or often disengaged reviews/meta-reviews. We suspect this is a meaningful contributor to much of the unpleasantness in ML peer review: **when recruited by force and with limited motivation and bandwidth, reviewers are likely less willing to go the extra mile (e.g., initiating internal discussions), as their main drive becomes fulfilling their assigned duties so their own submitted work is not desk-rejected.**

Perhaps recognizing this, many later conferences appear to be on the same page with us (in terms of realizing the negative side of mandatory reciprocal review): starting from ARR July 2025, technically qualified authors may request an exemption from review duty on a case-by-case basis if they find themselves lacking the relevant expertise. That said, broader allowance for exemption (e.g., lack of bandwidth) has yet to be widely adopted.

We find it ironic that a mechanism designed to distribute the workload ends up degrading its quality. Yet, if everyone could opt out with no consequences, the system would collapse under the sheer volume of submissions. The reasonable middle ground, then, is to allow sensible opt-outs while still holding authors accountable for their share of the reviewing load. We explore one such compromise in Section 4: a credit system that rewards good-faith reviewing and lets reviewers "spend" points to defer their reviewing obligations as they see fit.

### 3.4. Helpful Implicit Expectations of Actors Often Go Unmet

Like we previously teased in Section 2.2, conference processes implicitly assume that reviewers, ACs, and SACs will self-initiate best practices, such as timely calibration, substantive internal discussions, careful revision after rebuttals, and principled follow-ups, to ensure that informed decisions are made for each submission. We argue that in reality, **helpful practices like internal reviewer discussions rarely happen effectively at scale, because reviewers are seldom incentivized to "go the extra mile."**

These practices bring little recognition, credit, or accountability for the extra coordination and time they require, so under deadline pressure most actors would likely default to the minimum required. Steps like internal reviewer discussion thus tend to be perfunctory or skipped, and decision quality can degrade, not for lack of guidance, but for lack of aligned incentives to act on it.

## 4. Our Proposal: Fine-Grained Procedural Guardrails with a Currency-Like Incentive System

To make meaningful progress in peer review reform, we argue that two ingredients are essential: **enforceable procedural safeguards at different granularities**, and **an incentive structure that rewards good-faith participation while offering flexibility.** We propose a system based on a community-wide, cross-conference-supported economy called "OpenReview Points," mainly due to the widespread adoption of OpenReview and its good position to keep track of such balance.

This section outlines the basic principles of such a system, discusses potential enforcement strategies, and explores the feasibility of a conference-wide credit market that could finally provide conference organizers both the "stick" and the "carrot" they currently lack.

---

[4]Where such eligibility is often determined by prior publication records, such as number of published works at A* or similar conferences.

## 4.1. OpenReview Points: A Currency-Like Economy Enabling Flexible Options

The current review ecosystem operates on the honor system: a reviewer is expected to perform review duties diligently and hope that others will do so as well. However, we argue that optimistic hoping is not a system. To instill accountability, we propose a credit system, giving contributors to the review pipeline something to earn, spend, and track.

Under our proposal, ML practitioners would accumulate OpenReview Points based on their contributions to the community. For instance, in the context of reviewing, completing a standard review might earn 1 point, helping with an emergency review might earn 2, and being recognized as an "outstanding reviewer" could grant an additional 3 points.[5] Once earned, OpenReview Points could be spent to gain access to certain "perks" and privileges. For example:

- A reviewer can spend 5 points to opt out of an assigned review duty.
- An author can spend 10 points to exempt a co-author from their reciprocal reviewing obligation.
- An author can spend 50 points to request an additional expert reviewer in the case of a highly controversial or borderline decision. In the meantime, a reviewer/AC can take this job and earn those 50 points.
- An author can spend 100 points to redeem free registration.

This economy introduces direct incentives: if one contributes meaningfully, one gains flexibility and optionality. If one does not, one's publishing privileges will begin to shrink. It also gives conference panels more space to experiment with different policies and enforcement harshness, without relying on blunt-force policies like universal reciprocal reviewing or desk rejections.

For instance, much of our work argues that there is a lack of incentive for actors to "go the extra mile," even when such effort can be immensely helpful (e.g., as anecdotally demonstrated in Appendix C). With point incentives, however, such "extra miles" can be encouraged: reviewers may become more willing to initiate and engage in internal discussion, and ACs more willing to investigate, because exemplary actions can now be potentially rewarded. We can even push this further by introducing targeted awards and penalties to shape specific community behavior. As discussed in Section 2.2, the lack of a reviewer feedback loop can potentially be mitigated by awarding points to authors who provide detailed reviewer feedback that reviewers may con-

sult to improve their future practices. Similarly, as noted in Section 2.1, many reviewing issues stem from inflated submission counts. **One way to mitigate this is to require a small and refundable "submission fee" (e.g., 10 OpenReview Points) per paper.** If the paper is accepted or meets a reasonable "fair attempt" bar, the points are refunded; otherwise, they are forfeited. This soft deterrent discourages unready submissions by linking low-quality or premature work to a corresponding reduction in future publication privileges. We find this use of the credit system particularly appealing, as it directly targets the core issue of submission quantity. It also rests on a relatively reliable signal: the NeurIPS Consistency Experiments found that when two independent reviewer teams assess the same papers, their agreement is strongest on clear rejections. In other words, while review outcomes can be quite noisy for borderline or even spotlight-worthy work, they are far more consistent at flagging low-quality submissions, which is exactly what a refundable submission fee is meant to discourage.

While we have proposed several specific policies in this section, **we do not argue for the enforcement of any specific rule**, whether that is charging a submission fee in points, allowing opt-outs from reviewing, or anything else. Rather, we argue that a credit system would grant every participant in the ML community far greater flexibility in how they interact with the review process. **While we fully expect friction or disagreement regarding any particular rule or redemption policy, we believe it would be difficult to argue against the utility of having a credit system at all**, since it makes sense for different conferences to carve out their own rules to cater to their own communities.

## 4.2. Making the Credit System Enforceable: A Four-Part Defense

A credit system is only as useful as it is enforceable. Beyond awarding points for good-faith contributions, organizers need levers to deter gaming, such as fraud, point farming, and bulk low-effort reviewing. Realistically, these levers should operate at a finer granularity than the blunt instrument of desk rejection. No system is immune to abuse: even real-world economies with actual laws and tangible consequences face persistent bad actors. Nonetheless, we argue that, although imperfect, a credit system is far better positioned than the status quo, precisely because organizers can *award* and *deduct* points to impose countermeasures and realign incentives. We group these countermeasures into four reusable primitives that conferences can mix and match:

- **Duty Delegation:** restrict certain actions to specific roles. For example, only a paper's own (lead) authors may spend points to request an additional expert reviewer, and submission-fee-like charges fall on lead authors rather

---

[5] We emphasize that all point values mentioned in this section are intuitively assigned for hypothetical purposes. A real OpenReview Point-based economy would require significantly more sophisticated balancing, subject to each conference's own preferences. More on such specifications in Section 6.

than auxiliary ones. This blocks many forms of third-party interference.

- **Upper Limits:** cap how often an action may be taken per cycle, e.g., how many papers one may review for points, or how many additional or emergency reviewer requests one may file. This bounds low-effort point farming and bulk abuse.
- **Dynamic Pricing:** make repeated use of an action progressively more expensive, e.g., each submission beyond a threshold, or each successive additional-reviewer request, costs more points. This reserves scarce resources for the cases that matter most.
- **Voting-Based Penalties and Awards:** let peer reviewers and the AC issue fine-grained, peer-driven judgments, deducting points for low-quality reviews that peers and the AC confirm, and awarding points for exemplary ones (e.g., an "outstanding reviewer" bonus).

These primitives compose against concrete threats. For instance, bulk outsourcing of reviews is curbed by upper limits (fewer reviews per actor) and voting-based penalties (low-quality outsourced reviews lose points, making good reviews the easier path to rewards). Schemes that funnel points to a dedicated "point person" to be listed as a leading author across many papers are blunted by duty delegation and dynamic pricing, and, absent person-to-person transfers, are simply hard to pull off at scale. Spamming the "additional expert reviewer" option is also contained by per-cycle caps and escalating prices.

Crucially, much of this rests on one principle: **points must be earned through labor, not bought.** We therefore strongly advocate forbidding person-to-person point transfers and any money-to-points conversion. The moment points can be purchased, the system risks degenerating into pay-to-play, which is plausibly worse than the status quo. The voting-based lever in particular raises concerns about false positives and politicization, which we address in Section 5.2. We discuss further operational details, including governance and anti-fraud measures, in Appendix B.

## 5. Alternative Views

While we advocate for enforceable procedural safeguards and a credit system, we recognize that not everyone will agree with this approach. Below, we discuss several alternative perspectives and respond to their concerns.

### 5.1. "Peer review shouldn't be gamified."

A common objection is that introducing a credit system risks gamifying the review process, turning what should be a scholarly, community-driven responsibility into a transactional system. This concern is understandable, but peer review is already governed by incentives, such as review-

ing others' submissions in turn for having one's own work properly reviewed. It is just that these incentives are poorly aligned and implicitly stated.

Researchers submit to conferences because they care deeply about getting their own work accepted and to be featured on a platform. Yet, they are often disincentivized from reviewing carefully and consistently. A credit system does not create incentives out of thin air; it simply formalizes them and aligns them with the broader health of the ecosystem.

### 5.2. "Voting-based penalties will be abused or politicized."

The voting-based lever introduced in Section 4.2 naturally raises the concern that it could be misused, weaponized in borderline cases, or influenced by interpersonal bias. This concern is legitimate. Yet, as argued in Section 3.2, the reviewing problems that matter most are the subtle ones that desk rejection is too blunt to police, which is exactly why a finer-grained, voting-based penalty is needed: if the authors report a reviewer and that reviewer's peers, along with the area chair, agree that a review is unacceptably low in quality, the reviewer could face penalties ranging from a warning to graduated point deductions.

Such a system does introduce the possibility of false positives, but we argue this is largely acceptable. First, with guardrails such as close-unanimous (if not fully unanimous) agreement among the other reviewers on the paper, area chair confirmation, and an appeal mechanism, the practical false-positive rate can be kept low. Second, a point deduction is far less extreme than desk rejection or a submission ban, so even a wrong call is unlikely to cause severe or irrecoverable harm. Most importantly, the current system has the opposite problem: a nearly 0% true-positive rate, since no matter how badly a reviewer behaves, there are essentially zero consequences short of automatically verifiable atrociousness. The status quo is, in our view, worse in a comparative sense.

No penalty system will ever be perfect, but the absence of one leaves little room for improvement. We argue that a small risk of overcorrection is a worthwhile price for finally holding peer review to a higher standard. An even lower-risk alternative is to lean on the *award* side of the same lever, granting credits to ACs and reviewers who provide detailed feedback on their peers' reviews, enabling a positive feedback loop where actors have a channel to learn and improve. We discuss such recommended practices in Section 6.

### 5.3. "A credit system favors the privileged."

Some may argue that a credit system will disproportionately benefit researchers with more time, institutional support, or prior connections, allowing them to "buy" their way out

of responsibilities (e.g., being exempt from review duties) while leaving others to "pick up the slack." This is a legitimate concern, but in our design, points are earned through labor, not status. There is no "premium tier of citizen," only accumulated contributions through hard work. While it is still true that researchers with strong support will likely have more opportunities to contribute (as they are not otherwise occupied by some chores), their "surplus contributions" are still a net gain to the community.

We also note that, while exemptions from review duties might indeed result in losing reviewers, one thing to consider is whether those who are willing to pay a high price to be exempted are producing quality reviews (if kept by force), and whether they are likely to have the bandwidth to stay engaged with the authors. We tend to believe the answer leans toward the negative, and argue that a better alternative might be to just let them be exempted, and utilize the collected points to incentivize reviewers who do have the bandwidth and motivation in this particular cycle.

On the same note, one thing we would strongly advocate is **mobilizing researchers who are not main authors to participate more in the review pipeline**, as they likely have better bandwidth (since they are not under the pressure of author deadlines), and their reviews will not be as affected by feedback on their own submitted work. Under the current system, there is little incentive for researchers to do so, as most reviewers are recruited by mandatory reciprocity, which no longer applies without being an author. Our proposed credit system might provide them with a strong incentive to participate, as they can earn points to enrich their publication privileges; and specifically, have the option to spend such points to be exempt from reviewer duties when they are submitting lead-authored work, granting themselves wider bandwidth as authors when they are under rebuttal pressure.

### 5.4. "Restricting point transfers is undemocratic."

It is worth first being explicit about why unrestricted transfer would break the system. Points are meant to certify that a specific person performed a specific service; once they can be gifted or sold, a researcher's privileges no longer reflect their own contributions, and the "labor for perks" principle the whole design rests on (Section 4.2) loses its meaning. Transferability would also neutralize much of our four-part defense: the per-cycle Upper Limits and escalating Dynamic Pricing both assume points are tied to an individual, so a well-resourced actor could simply pool points from others to exceed those caps and absorb the rising costs. A transfer channel is, moreover, the natural on-ramp to a black market, where large groups funnel points to a few "point persons" and points are quietly sold off-platform, precisely the money-to-points, pay-to-play outcome we most want to avoid, and plausibly worse than what we have today.

Given this, one might still object that strictly regulating transfers is paternalistic, much like tightly regulating the transfer of wealth. We find the analogy imperfect: points are not private wealth but a record of personal community service, closer to a professional credential, a citation, or an authorship than to money. One cannot sell a degree or a reviewing record, and few would call that undemocratic; non-transferability is what gives the credential its meaning, not a liberty taken away. We also want to note that the restriction on person-to-person transfer is not a blanket ban on all sharing/pooling operations: as noted in Appendix B, team-based redemptions (e.g., coauthors jointly spending points) remain sensible, and each conference can set its own boundaries. What we resist is decoupling points from the labor that earned them.

### 5.5. "All of this sounds too bureaucratic."

Some may worry that procedural enforcement, tracking points, and adjudicating review quality will introduce too much bureaucracy into the process. This concern, too, is understandable. However, conferences already invest massive effort coordinating thousands of reviews and rebuttals; we are simply proposing mechanisms to make those efforts fairer, more consistent, and more sustainable over time. While we acknowledge that the full form of our credit system can be too heavy to be implemented at once, we argue a gradual rollout of changes can be a rather "soft landing" to existing community members, as we detail in Section 6.2.

### 5.6. "Will points simply inflate as the system scales?"

A natural concern at scale is not the raw submission count but the points themselves: as the system runs across many cycles and venues, the total points in circulation could grow until they inflate and lose value as either a deterrent or a reward. This is a familiar problem for real-world point economies, and we can borrow their remedies. Credit-card and loyalty-point programs routinely curb inflation and hoarding through point expiration and time-limited, discounted redemption windows that nudge members to spend rather than stockpile, giving organizers direct control over the effective point supply. A second, related worry is that the system could be gamed at scale through cheap, bulk, or AI-driven point farming; this is contained by the four-part defense of Section 4.2, where per-cycle caps limit low-effort accumulation, dynamic pricing makes repeated requests progressively more expensive, and peer/AC voting penalizes the low-quality output such farming would produce (see Appendix B for further operational notes). None of this makes the system immune to abuse at scale, but it does hand organizers concrete moderation levers that current conference mechanisms simply lack.

### 5.7. "A cross-conference reciprocity must exist first."

One clear and legitimate criticism of our credit system is that, for it to work to its full potential, multiple major conferences must adopt it. Granted that conferences like ICML, NeurIPS, and ICLR almost never work together closely, we recognize that such a prerequisite can be difficult to achieve. However, we argue that there are ML conferences well-positioned to adopt such practices: e.g., the ARR series of conferences has long implemented cross-conference measures (e.g., submission bans from the next ARR cycle), as experimented with in EMNLP 2025, making them more openminded to adopting other similar cross-conference measures. Further, even if the credit system is per-conference, it can still function at a level that is better than nothing; it is just that features requiring accumulated effort may be harder to activate and experiment with.

### 5.8. "What about early-career researchers?"

Much like how games onboard novice players and companies onboard new employees, the point-hosting platform could grant a baseline amount of points to first-time contributors, perhaps along with a protection period (much like how new hires are not immediately subject to performance-improvement or dismissal procedures), giving them enough time and capital for trial-and-error. Concretely, such a protection period (e.g., the first 3–5 months, or one's first conference cycle) might allow a limited number of initial submissions to be fully refunded and non-extreme penalties to be softened, so that newcomers can learn from early missteps without those missteps derailing a nascent career.

We further note that a credit system can reward "good acts" well beyond reviewing itself. For instance, conference organizers might host onboarding workshops, or pair early-career researchers with more experienced "research buddies" who mentor them on submission compliance, review writing, and community norms. Rewarding such mentorship may be far more constructive than relying on point awards and penalties alone, and would be hard to establish without a fine-grained credit system in place.

### 5.9. "Who pays for the cost?"

Our mechanism does not require conferences to offer more complimentary registrations than they already do. Major ML conferences already grant free registration to top reviewers; NeurIPS 2025, for instance, offers this to an estimated 1,900+ individuals. With conferences like ICML becoming more open to virtual attendance, the material cost per registration is further reduced. As discussed in Section 6, point thresholds can be calibrated to match existing resource constraints. The key difference is simply how these perks are allocated: instead of relying on AC discretion, we rank reviewers by earned credits, providing a more objective way to identify top contributors.

### 5.10. "Just enforce monetary cost per submission."

Some ML conferences have begun experimenting with monetary submission fees to curb submission volume; IJCAI-ECAI 2026, for instance, charges $100 per paper from the second submission onward. We argue that points are preferable to money: if the monetary threshold is too low, it becomes meaningless as a deterrent; if too high, it excludes researchers without strong institutional support. Points, by contrast, can be earned by contributing to the community (e.g., by reviewing papers), making them a fairer currency that rewards effort rather than financial privilege.

## 6. Recommended Practices / Call to Action

As emphasized throughout this position paper, our goal is not to promote a single, prescriptive rulebook that every conference must follow, but to advocate for a flexible framework that can adapt to different conference idiosyncrasies. Every policy comes with its own trade-offs; it is, ultimately, an art of compromise to decide which set of policies to adopt. **Under our credit system, conference panels and authors are akin to store owners and customers: the panels decide what goods are offered and at what price, while the customers decide where and how they wish to spend their money.**

However, we fully recognize that without concrete discussion of how such a system might actually operate, there could well be resistance and, worse, chaos if adopted without proper consideration. Thus, this section serves as practical guidance from us authors on how the first few conferences adopting a credit-like system might proceed. We also leave our answers to many frequently asked questions to Appendix B.

### 6.1. Heavy on existing perks

We believe that early adopters should anchor their credit-like system around existing perks already offered in current ML conferences (e.g., complimentary registration, emergency reviewer invitations), rather than immediately introducing entirely new perks (e.g., exemption from review duties, invitation of extra reviewers). Staying with existing perks provides two immediate benefits: 1) Because these perks are already part of established workflows, redistributing them according to the credit system (e.g., ranking reviewers by awarded points per conference cycle rather than relying on an AC's subjective judgment) keeps overall impact bounded. If the new distribution turns out problematic, its effects are still confined to the known scope of these already-tested perks; whereas brand-new perks introduce unknown

risks. 2) We can directly compare key metrics between credit-system conferences and their historical data. Any improvement or decline is then more likely attributable to the credit system (or its specific implementation), rather than being confounded by the introduction of new perks. Using the same set of perks but with credit-based allocation gives us an opportunity to collect ablated data on how the credit system behaves in practice. Such data shall serve as the baseline to test out different new perks.

## 6.2. Gentle and gradual rollout of new perks, potentially with one-off tests

When launching new perks, it is best to roll them out gently and gradually rather than all at once. This approach offers a clean testbed to monitor each perk's contribution and reduces the information load on all involved parties, who will need time to adjust.

Observant readers may notice that some of our proposed policies (e.g., free registration, the right to request additional reviewers, or refundable submission fees discussed in Section 4) require a relatively long-term accumulation of points to become useful. Researchers would need to earn (or be penalized) enough points before crossing the thresholds of these services, stretching the evaluation horizon.

A simple way to expedite early evaluation is to introduce one-off tests. For example, in addition to point awards, conferences might grant top point-earners a one-off right to request an additional reviewer to help resolve borderline cases, with this right expiring at the end of the conference cycle. This allows organizers to directly observe whether such redeemable incentives meaningfully help, and to what extent these improvements propagate through the reviewer–AC pipeline. This kind of fast feedback might help conference organizers trim unhelpful policies quickly and expedite faster iterations of rule sets.

## 6.3. Determining point values for contributions and perks

One reason we did not specify exact point values for different contributions (e.g., how many points an emergency review should yield) is that we currently lack the empirical data needed to set these responsibly. As a general guideline, we believe it is reasonable to treat the completion of one regular review duty as the base "unit price" of this ecosystem.

Rather than arguing directly about how many points each contribution "should" receive, we propose working backwards from the perks: estimate how many free registrations or similar rewards a conference can offer, determine what percentile of contributors this represents, and calibrate point values accordingly. Detailed considerations are discussed in Appendix B.

## 6.4. Track key metrics and publicize such statistics

Finally, for a credit system to have a lasting impact, conferences must make informed decisions about which rules to adopt and at what point-values. Such decisions require cross-conference consistency. If, for example, NeurIPS values its perks at 10x ICLR's level for no meaningful reason, the ecosystem loses the interoperability we envision. Thus, each conference should monitor key metrics and publish these statistics as part of their post-conference fact sheets.

For example: If a new rule is implemented, do we observe increased interaction among reviewers and ACs? Do ACs report that these additional exchanges help them make more confident decisions? These statistics and reports can form the foundation for iterating toward a better implementation of the credit system, and can serve as a strong signal, even an advertisement, encouraging more conferences to adopt a shared credit currency.

## 7. Conclusion and Limitations

In summary, we present a flexible credit system aimed at improving accountability, aligned incentives, and procedural fairness in the ML peer-review pipeline.

A proposal is only faithful if it also highlights its main weakness. In our case, it is **the lack of numerical experiment results.** However, we believe discussions about review mechanisms are most meaningful in the hypothetical space, since there is no way to rewind history and A/B-test two parallel conference processes. LLM-powered simulation is a natural alternative, but at the scale of multiple conferences, it adds little value: too many layers of prompt, decoding, and model-specific variance compound, making outcomes highly sensitive and difficult to trust over such long horizons. A further complication is that many key metrics, such as the number of internal reviewer discussions or reviewer-profile dynamics, are simply not public. These restrictions make even crude simulations hand-wavy: changing a threshold or banning simulated authors from a "next conference" would trivially reduce load, making our proposed mechanism "win," but such results would be meaningless without real baselines. That said, we recognize the desire for some anchoring to real conferences. **Thus, we share three case studies (drawn from top ML venues where we served as reviewers) in Appendix C for a compromise between realism and faithfulness.**

Last, we disclose that part of the writing of this paper was polished by a language model, though a human researcher is there to verify that the final output is true to the researcher's opinion.

## Acknowledgements

We thank our reviewers for their concrete and constructive feedback, and in particular for the many interesting threat models they raised, which shaped the four-part defense of our credit system. Due to page limitations, we could not include a visualized diagram of our proposed pipeline, but we will make sure such a visualization is available at our poster.

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

# A. Related Works

**Proposal of conference mechanisms for better ML review quality**    To the best of our knowledge, few published works have addressed **how to improve ML review quality by proposing new conference mechanisms.** The closest work to ours might be Kim et al. (2025), where the authors advocate establishing a feedback loop to reviewers and promoting reviewer rewards — two points that we also share. Specifically, Kim et al. (2025) highlights that if we allow authors to rate reviewers, those ratings will almost always be heavily influenced by the specific strengths and weaknesses (and, by extension, the scores) listed by the reviewer (Goldberg et al., 2025). Reviewers who fairly rate papers negatively may be subjected to unfair retaliatory ratings from authors. To address this, Kim et al. (2025) suggests a two-stage reveal: authors first read only the reviewer-written summary and strengths and provide a rating, after which the weaknesses are revealed. We argue that this system might work to some degree, but in reality, much of a review's quality is determined by whether the highlighted weaknesses are sound and well supported. Rating reviewers without seeing these details would likely produce noisy signals, and reviewers would be incentivized to write vague summaries that lack sharp substance; even worse, reviewers could inflate their strengths sections to manipulate the rating. The Kim et al. (2025) authors are even transparent in admitting their proposed mechanism cannot address low-quality negative reviews,[6] which are likely the main complaints of most authors.   As for reviewer rewards, Kim et al. (2025) mostly argues for vanity perks like digital badges (e.g., ones similar to the "Pull Shark" badge on GitHub). We argue that such vanity-only perks would have much less influence than the review, submission, and cost-influencing perks we propose here.

Another piece of related work is the Isotonic Mechanism score pioneered by Su (2021) and its follow-up works like Wu et al. (2023); Su et al. (2025), which survey authors (with multiple submissions) and ask them to rank their submitted papers. A score is thereby calculated and compared with the mean of reviewers' raw scores. Should these two scores exhibit too drastic a gap, it may trigger AC intervention with an additional reviewer request, etc. We note that this work is, by and large, orthogonal to ours, as it is yet another safeguard one can implement under our proposed credit system. The two works overlap in the sense that some countermeasures do show resemblance (e.g., requesting an additional reviewer).

A broader but still relevant work is the survey by Shah (2022), which systematically examines peer review challenges across several dimensions: mismatched reviewer expertise, dishonest behavior, miscalibration, etc. For each dimension, Shah discusses computational solutions — such as randomized assignment algorithms to mitigate dishonest bidding, or machine learning approaches to address commensuration bias. While comprehensive, the survey proposes separate solutions tailored to each problem rather than a unified mechanism. Our credit system, by contrast, offers a single flexible framework that can incorporate many of these solutions as special cases.

Two other pieces of related work are Rogers & Augenstein (2020) and Zhang et al. (2022). Rogers & Augenstein (2020) discusses why incentives clash under a peer-review context and outlines many potential proposals (e.g., better review–paper matching, more tracks, abolishing "score-based" feedback, track-specific review formats, etc.). Similarly, Zhang et al. (2022) investigates various policies but focuses on modeling why resubmission is so prevalent despite many works eventually getting accepted at a top venue. Like Shah (2022), these works propose custom solutions for each problem or edge case, rather than a unified recipe.

**Conference Review Statistics**    There are a few works like Yang (2025); Cortes & Lawrence (2021); Beygelzimer et al. (2023); Goldberg et al. (2025) that collect real statistics and conduct controlled experiments from past conferences. While such works typically do not propose mechanism-based solutions, their numerical presentations help illustrate the scale and muddiness of current conferences; yet, such controlled exploration shall offer us insight into the practical dynamics of a particular mechanism design.

**Broader Relevant Art**    Last, outside the machine learning community, we have Gasparyan et al. (2015) analyzing peer review incentives under a mainly medical-focused context. The main argument of this work is *"none of these (financial or nonfinancial incentives) is proven effective **on its own**"*; however, the authors envisioned *"a strategy of combined rewards and credits for the reviewers' creative contributions seems a workable solution."*

Our work makes essentially the same argument, though framed under a credit system (and of course, with more ML-specific flavors). Our system supports many of the typical financial and non-financial incentives mentioned in Gasparyan et al. (2015). For instance, several exemplary policies we discuss in Section 4.1 range from incentives for reviewers to write higher-quality and more timely reviews, to deterrents for authors submitting unready work, to non-financial privileges such

---

[6]https://openreview.net/forum?id=l8QemUZaIA&noteId=SXdGgGs6SV

as the right to be exempt from review assignments, and even financial compensation like free registration. We believe it is fair to say that we are not advocating for any particular type of incentive, but rather a collection of them. It just so happens that we unify them under a credit system, allowing their impact to last beyond a single conference.

Another point specific to Gasparyan et al. (2015) is that many reviewers dismiss incentives such as paper purchase discounts or free publication access, since their institutions already pay for publisher subscriptions, making such incentives mostly relevant to members with non-academic backgrounds. In our proposal, however, conference hosts can offer services that no institution would purchase in bulk (e.g., registration), or even privileges that money cannot buy (e.g., the right to request additional review resources). While we do not claim these incentives are inherently better — as one man's vulgarity is another's lyric — we do believe our framework offers a flexible way to combine different incentives to suit diverse needs, effectively pushing forward a system that Gasparyan et al. (2015) envisioned.

Related ideas also surface outside academia, in the opensource community. For instance, `vouch`[7] implements a "web of trust" in which members vouch for (or denounce) one another to manage participation. While conceptually adjacent to our credit system, the resemblance is loose: `vouch` is largely front-door-focused and access-control-oriented, gating who is admitted in the first place, whereas our proposal is aimed at shaping behavior after admission (for obvious fair-science reasons), with greater flexibility in how earned points can be spent. We thus see it as related in spirit but distinct in goal.

Outside the effectiveness of a certain incentive system, much of a conference's experience also depends on many other aspects, such as the reduction of collusion rings (Jecmen et al., 2025), finding better reviewer–paper mappings (Mimno & McCallum, 2007), determining the level of openness (Rao et al., 2025), or the use of LLMs for reviewers (Liu & Shah, 2023). We refer users to such works to build a deeper understanding of these important topics. We recommend readers refer to Kim et al. (2025) for an overview of such works.

---

[7]https://github.com/mitchellh/vouch

# B. Frequently Asked Questions

**Who governs this?** We believe it is best to have a bank-like entity to govern point storage. Neutral platforms like OpenReview could serve as an ideal medium for tracking how many points each author has accumulated.

**How do we prevent bad behavior (fraud, point farming, unfair allocation, etc.)?** No mechanism can fully eliminate misconduct. Even real-world economies with tangible consequences face persistent bad actors. It would be disingenuous for us — or anyone — to claim otherwise. However, any thriving economy depends on aligning the interests of its participants. Conference organizers and authors share the goal of obtaining high-quality reviews and meaningful discussion, while reviewers (in the worst case) may simply want to earn points.

To bridge these goals, additional points can be awarded for high-quality reviews, and penalties applied for low-quality ones. We can also limit the total review capacity per author to discourage "quick review for basic point farming." Enforcement can be driven by an AC + reviewer voting mechanism, as discussed in various parts of our paper. We believe that these enforceable safeguards would be effective in mitigating typical point-farming behaviors where review quality is severely compromised.

**Would Y operation be allowed (e.g., point transfer)?** Point transfers should be rare and limited, as the foundation of this system is that contributors must personally engage in community service to earn points. However, some team-based privileges make sense. For instance, authors of the same paper could collectively "purchase" certain "products" — e.g., request additional review resources for borderline cases, or exempt a non-expert coauthor from review duties.

In line with our goal of proposing a framework rather than a detailed rulebook, we find it reasonable to forbid person-to-person transfers while allowing team-based purchases or awards, depending on conference preferences.

**Would Z infra/consensus be needed?** It is fair to note that certain infrastructure and shared consensus would be required to operationalize our credit system. At minimum, platforms like OpenReview should provide:

• A balance-tracking system to record point accumulation per author.
• A limited point-transfer interface to implement awards or penalties.

While such infrastructure is necessary, we emphasize that it is a minimal requirement. The "legislation" and enforcement of specific policies would still rest with individual conference panels. As one man's vulgarity is another's lyric, there will never be a perfect policy — only carefully considered trade-offs. Our framework respects the diverse needs of the community and supports them with great flexibility.

## C. Three Case Studies Where We Serve as the Reviewers

While our position paper, unfortunately, lacks real conference data to support why our proposed framework would be helpful, we believe there is even less reason to run an LLM-roleplaying simulation. We understand the perspective that having some anchors to real conferences is preferred. Here, we share three case studies — all from similar top ML conferences — where we serve as reviewers.

### C.1. Case 1: Reviewers criticizing matters outside the paper's scope.

In this case, we observed that another reviewer criticized the submitted work for reasons clearly outside its intended scope. We therefore raised our concern to the AC:

---

**Internal comment to PC/SAC/AC**

This message is set to be only visible to PC, SAC, AC, and the authors.
I want to disclose that I find reviewer `A`'s evaluation of this paper quite unreasonable. This reviewer writes:

- Certain methods, such as `method type`, demonstrate limited effectiveness in `setting`, which may restrict their practical deployment.

- The paper points out that many of the `an important task` methods tested are essentially extensions of existing models adapted for `another important task`, such as `a famous method`.

It doesn't make much sense to cite the low performance of certain featured methods as weaknesses of a dataset-proposing/benchmark paper. It is not the authors' problem if an established method underperforms. Instead, the point of benchmarking is precisely to show when a method would fail. Many benchmark works have done this — `some examples` — and it is incomprehensible why this is considered a weakness.
Another criticism from reviewer `A` is:

- The tasks within the benchmark may not capture all possible real-world application scenarios, possibly overlooking specific needs within certain domains.

This, in my opinion, is a boilerplate concern that can be said for literally *any* dataset. While I do agree that the proposed dataset does not capture some important `task` scenarios — `some examples` — criticizing it for "not capturing all possible real-world applications" crosses the line and feels borderline hostile. This is akin to criticizing a method paper for not evaluating on every possible dataset.
I recommend the AC to either disregard `A`'s review or consider encouraging the reviewer to revisit the evaluation.

---

This paper was ultimately accepted. This anecdote shows that without inner-reviewer analysis, simple rule-based policies such as "inactive → desk rejection" fail to capture cases of severely low-quality reviews. Finer-grained measures must be practiced to ensure that positive impact scales broadly, rather than being limited to a few desk-rejected papers.

### C.2. Case 2: Reviewers asking for particular experiments after the rebuttal deadline.

In a top ML conference where the exchange between authors and reviewers is limited to a certain time window, we had a split decision situation where the non/late-responding reviewers are not supportive of the submission. As the AC is calling for consensus, we jumped in and asked:

---

**Internal reviewer discussion**

I skimmed over the two negative reviews of this work and found merits in many of the reviewer-raised points. However, I also find the authors' rebuttal to be proper in many regards — especially when the raised concern demands a clarification-like answer.
It looks like the two negative reviewers have yet to address the authors' rebuttal in a meaningful way (only acks are issued, cmiiw). So, to reach a consensus, I believe it would be helpful if the two reviewers could elaborate a bit on their leftover concerns. I am happy to set aside some time to discuss such leftover issues from my perspective.

---

Essentially, the two negative reviewers believed that certain experiments were missing — one of which can be seen as a combination of two existing methods, and another as a specific investigatory study of the author-proposed method. While we find such suggestions to have merit, we believe they were not raised appropriately from a procedural standpoint:

---

**Internal reviewer discussion**

I appreciate `A`'s detailed response and updated review. I believe **A (as well as B)'s main concerns regarding `ProposedMethod` vs. `PriorWork1 + PriorWork2` are legitimate and sound.** That being said, I am always the kind of reviewer who is "more in the authors' shoes"—for lack of better words—and I would like to present two alternative arguments regarding this concern.

First, I believe experiment-comparison requests that touch on *combinations of existing works* should be cautiously brought up. Many methods can be combined, but their combinations typically require a number of discretionary design decisions, and it is often unlikely for authors to feature the exact combination a reviewer has in mind. In this case, `ProposedMethod` proposes a paradigm of `[redacted]`, where the scope of eligible combinations is wide. Thus, in my opinion, **if reviewers are specifically interested in the comparison `ProposedMethod` vs. `PriorWork1 + PriorWork2`, such a request should be made *explicitly* before the rebuttal deadline, rather than mentioned in hindsight when the authors have no channel to address it.**

From the look of it, the `ProposedMethod` authors submitted their initial rebuttal on `an early date`, which is `[redacted]` days after the review post. However, only `A` engaged substantively on `a late date`. I must note that this year's `BigConferenceName` requires only two rounds of exchange, yet only 2/4 reviewers provided those to the authors, with all engaged reviewers leaning positive. **For such reasons, while I am also interested in this comparative result and agree with A's analysis, I do not believe we can use it against the authors (at least not as a singular veto reason), as the request was not properly raised from a procedural standpoint.** Imagine we were submitting a paper where reviewers were largely non-responsive, and the paper was then rejected for missing an experiment that was never explicitly asked for—it would be hard not to feel that is unfair. While I understand we all have different priorities and may have limited bandwidth for various reasons, we need to properly compensate authors when such situations occur.

(I know it is uncommon for a reviewer to argue on behalf of authors, but I always do so when it is warranted. The AC is welcome to confirm that this advocacy is from me for good reasons and not the result of any collusion.)

---

Later, we and the two reviewers exchanged more than five comments in total, which helped the AC reach a favorable decision. This anecdote shows that many reviewers are willing to engage in internal discussions, and such exchanges are profoundly helpful in deciding borderline papers — they simply never had the "push" to initiate such discussion voluntarily. We argue that our credit system could help elicit more of these productive discussions.

Another takeaway from this exchange is the importance of having enforceable safeguards (e.g., reply deadlines), as otherwise authors may have no channel to meaningfully rebut at all. Attaching rewards and penalties to such actions is also crucial to ensure procedural fairness. While we respect and appreciate `A` and `B` for engaging our discussion, the fact that they were ok with not replying / late replying authors is because there is virtually no penalty to them (as their "misconduct" would be too minute for desk rejection, yet the conference organizer has no other finer-grained tools) — something our credit system would help address.

**C.3. Case 3: Authors present unsupportive results while claiming otherwise.**

In this case, we found that the authors were presenting unsupportive results to one reviewer while verbally claiming the opposite — so we stepped in:

---

**Internal reviewer discussion**

As another reviewer, I find the reading on `ProblemX` tricky. The goal of `Task` is to have the `[redacted]`. In `ProblemX`, when the `component` is trained only on `dataset1`, the `dataset2` accuracy drops quite significantly — a disadvantaged result. Once the `component` is trained on both `dataset1` and `dataset2`:

- The `dataset1` performance improves `a small number` over the `a baseline`, but this also comes at the cost of generating many more tokens than the `a baseline`.

- `[redacted as it is too technical]`

- In the end, the system shows only a `small number` accuracy gain on the task it was specifically trained on, while `doing something at a higher cost`. I think the added experiment makes the work more negative (though I appreciate the transparency), as it feels like the proposed `component` is very task-specific and does not generalize well.

(This message is set to be not visible to authors so that we can have a discussion with supposedly no bias, but I am happy to adjust the visibility should you want a more public discussion.)

---

The reviewer exchanged views with us and we reached an agreement. The paper was ultimately rejected, with the AC explicitly citing our reasoning. Specifically, this other reviewer noted:

---

**Internal reviewer discussion**

(BTW, you're among the very few reviewers who respond to other review comments. I truly admire this level of dedication.)

---

This suggests that internal reviewer discussion is indeed rare, even though we lack direct statistical evidence.

We hope these three anecdotal case studies illustrate how even a small component of our proposed framework — encouraging more internal reviewer discussion — can meaningfully facilitate paper evaluation. While we acknowledge our bias, we believe that in all three cases, the key factor behind each paper's acceptance or rejection was our initiative in starting those discussions. This suggests that most reviewers *want* to contribute and most ACs will take such discussions seriously; they simply need a small push to take the initiative — a push that our credit system could very well provide.

