# OpenReview forum: "Position: Want Better ML Reviews? Stop Asking Nicely and Start Incentivizing with a Credit System"
_ICML.cc/2026/Position_Paper_Track — ICML 2026 Position Paper Track regular_

### Official Review · Reviewer_Gc4s · 2026-03-03

**Significance:** 3
**Argument Clarity:** 3
**Rating:** 4
**Confidence:** 3

**Questions:**

1.Could you consider introducing some more specific and verifiable simulation framework designs? For instance, based on existing conference data (such as reviewer response rates, distribution of review quality, and submission growth curves), could you quantitatively model potential behavioral changes (such as the proportion of high-quality reviews, frequency of internal discussions, and changes in submission count) that may occur after introducing this system?

2.Section 5.6 mentions providing "start-up credits" and "protection period" for the early-career researchers, which is a promising direction. However, how to implement it specifically? For example, how should the number of "start-up credits" be set to balance encouraging participation and preventing abuse?

3.The credit system may give rise to new forms of conflicts of interest. For instance, could a senior researcher with a large number of credits potentially use his influence (such as threatening to use his credits to request additional reviews of competitors' papers)?

4.How does the AC/PC assess whether a review is "high-quality" when allocating reward credits? How can favoritism be avoided?

**Alternative Views Section:**

Yes

**Compliance With Llm Reviewing Policy A Conservative:**

Affirmed.

**Discussion Potential:**

2

**Final Justification:**

The rebuttal addressed my main concerns,  and I agree with the acceptance of this paper.

**Paper Summary:**

This paper aims to address the shortcomings of the peer review system in the machine learning community. It indicates two core problems existing in the current review system: (1) the explosion of submission count; (2) the absence of incentive mechanisms. The core solution is proposed by the authors, i.e., enforceable yet fine-grained procedural safeguards, supplemented by a cross-conference currency-like credit system called "OpenReview Points". This paper not only presents its own ideas but also thoughtfully discusses alternative views.

**Position:**

Yes

**Position In Title:**

Yes

**Related Work:**

3

**Strengths And Weaknesses:**

Strengths:

1.At the beginning of the paper, the authors clearly points out that the dilemma of ML review cannot be resolved through "polite requests" or "optimistic guidance". Then, they systematically analyzes the two core challenges of "submission inflation" and "lack of incentives", and critically evaluates the limitations of existing attempts (such as soft and hard submission caps, 100% mandatory reciprocal reviewer recruit, etc.),

2.The proposed "OpenReview Points" credit system is the core contribution of this paper. This system monetizes review contributions, allowing researchers to "earn" points through good reviews and "spend" points to obtain privileges (such as requesting additional reviews).

3.The paper discusses several alternative perspectives and addresses concerns such as "gaming", "abuse of voting-based penalties", "exacerbation of privilege inequality", "excessive bureaucracy", and "the need for cross-meeting reciprocity".

4.The "Recommended Practices / Call to Action" section provides operational suggestions, such as gradually introducing new privileges based on existing perks, deriving point pricing from benefit value, and tracking and publicizing key metrics. These suggestions significantly enhance the feasibility of the proposal.

Weaknesses:

1.Although the paper discusses the concern about "gaming" in Section 5, a carefully designed point-based credit system may trigger unexpected complex game behaviors in practice (for example, deliberately seeking "emergency review" opportunities to earn more points). In the Appendix, some preliminary defensive designs can be proposed to tackle potential high-risk game scenarios.

2.This paper proposes a system process. Adding a conceptual architecture diagram (showing the relationship between researchers, submissions, reviews, point earning/consumption, and conference policies) and a timeline of the implementation roadmap can greatly enhance readability and comprehension of readers.

3.The main text employs multiple expressions with similar meaning, such as "currency-like credit system", "OpenReview Points", and "point-based system". It is recommended to maintain consistency throughout the entire text to enhance professionalism.

**Support:**

2

---

> ### Author Rebuttal · Authors · 2026-03-27
>
> We thank the reviewer for the feedback and support.
>
> ## **`W1, Q3&4 - Add defensive design in appendix` We already have some, but here's a defensive system.**
>
> We shall be the first to admit our system can be gamed. Even in the real world, there is no shortage of financial criminals. We do not pretend our 8-page position paper is ironclad, and we are explicit about it in `L077L`, `L266R`, `L610`, `L636` and more.
>
> In fact, one may view the unpleasantness of current ML review a product of gaming, where irresponsible actors are not doing their job, since that's enough get pass. We argue that, **though imperfect, the credit system has a much better fighting chance: by awarding/deducting points, organizers can impose countermeasures and motivate good acts.**
>
> In `App B: FAQ - How do we prevent bad behavior (fraud, point farming, unfair...)?` we introduce some countermeasures. Here, we expand it into a four-point defense:
>
> 1. **Duty Delegation** — restrict actions to specific roles.
> 2. **Upper Limit** — cap how often an action can be taken.
> 3. **Dynamic Pricing** — increase cost with usage.
> 4. **Voting-based Penalties/Awards** — enable peer-driven rulings.
>
> For the reviewer-raised "farming emergency reviews" concern, #2 shall directly help by imposing a hard cap on how many emergency reviews one can claim per cycle. The existence of #4 may also act as a deterrent, where the reviewers might hesitate to write many subpar reviews in fear of penalties, and are motivated to write good ones for the reward potential.
>
> Similarly, for your `Q3`, #1 shall require that only the authors of a paper can spend points to request additional expert reviewers. This removes the possibility of third-party interference. We can also use #3 to make such requests increasingly more expensive, so that the authors are more motivated to spend points on their most worthy submissions.
>
> Similar principles can help partially alleviate the `Q4` AC favoritism concerns. Instead of AC recommending top reviewers or purely by number of reviews done (current practice), good reviews can be voted out by peer reviewers per #4. We can also make such metrics accumulative — e.g., one needs to provide multiple good reviews to receive "good reviewer awards," which naturally spans multiple papers / ACs, reducing intentional favoritism.
>
> ## **`W2 - Make a diagram` Good idea, here's a sketch!**
>
> Here is [an ASCII diagram sketch](https://anonymous.4open.science/r/credit_ml_review-E201/README.md), this is char limit hungry, so we leave it to an anomyous repo.
>
> ## **`W3 - Term consistency` Will do!**
>
> We will consistently use *"credit system"* to describe the system and *"points"* to describe the trading medium. Thanks to highlighting this.
>
>
> ## **`Q1 - Verifiable simulation designs` We respectfully believe a faithful simulation is not practically feasible nor scientifically meaningful. Please hear us out.**
>
> While simulation is a reasonable request for a future-envisioning paper, we opt to respectfully push back here for three main reasons:
>
> 1. Review is dynamic and knowledge-intensive; LLM are not yet capable of faithfully simulating that many moving factors.
> 2. The credit system is accumulative, which calls for multi-conference simulation, which shall introduce huge compounding error.
> 3. Key metrics (e.g., internal discussion rate, emergency review requests) are not public or not standardized (e.g., review quality), so there lacks baseline.
>
> Thus, simulation would be largely hand-wavy. For example, penalizing "bad authors" so they have no points left would trivially reduce load and make our system "win." But such results lack meaning, as it is easily steered and without clear real-world baselines. Prior work on review mechanisms typically does not rely on simulation. Instead, real-world pilots are used (e.g., NeurIPS' consistency, ICML's self-ranking, IJCAI's submission fee). We view this as the appropriate path.
>
> That said, we understand the desire to have better practical connections. To anchor our claims, we include three case studies (App C). While anecdotal, they show that incentivizing small actions (e.g., initiating discussion) can materially improve outcomes.
>
> ## **`Q2 - Start-up specifics` Good Q, here are some thoughts.**
>
> Our system introduces *penalties*, which may burden early-career researchers. Thus, we propose start-up credits and a protection period (e.g., 3-5 months), during which a limited number of initial submissions can be fully refunded, and non-extreme penalties can be reduced. In such way, the lessons are still learned without destroying one's career.
>
> We also believe one good use of credit system is rewarding "good acts" even beyond the context of paper review. E.g., conference organizers may host workshops or form "research buddies," where experienced scholars can mentor undersupported early-career ones to pick up necessary skills such as submission compliance, review writing, etc. These can be more helpful than taking/giving points.

---

> > ### Author Rebuttal · Reviewer_Gc4s · 2026-04-01
> >
> > Thanks for the detailed response. I will hold the rating.

---

### Official Review · Reviewer_Fd3q · 2026-03-09

**Significance:** 3
**Argument Clarity:** 3
**Rating:** 5
**Confidence:** 3

**Questions:**

No questions.

**Alternative Views Section:**

Yes

**Compliance With Llm Reviewing Policy A Conservative:**

Affirmed.

**Discussion Potential:**

4

**Final Justification:**

All concerns were covered.

**Paper Summary:**

The authors propose a credit system for peer reviews. Their position is that peer review in machine learning is unlikely to improve through polite requests or optimistic guidelines in calls for papers. The paper proposes a discussion related to two central questions:
1. How can we reasonably limit the number of submissions?
2. How can we incentivize good review practices and discourage bad ones?

To address these questions, the authors propose two methods: (i) fine-grained procedural safeguards and (ii) Openreview points. Their aim is to make peer review failures in machine learning less frequent and more sustainable.

**Position:**

Yes

**Position In Title:**

Yes

**Related Work:**

4

**Strengths And Weaknesses:**

The paper is very well written, argued, and makes quite clear contributions. The objective is well stated, and the authors' position is clear from the moment you start reading the article. It describes a very prevalent problem in the machine learning community. This problem has been growing for some years now, and the authors propose an innovative and interesting way to address it. I personally believe it can spark a lot of discussion that will benefit the community.

Furthermore, the problem is well described and argued because the authors identify its main causes. The first is that the high volume of submissions implies different types of complications and challenges. The second main problem is a lack of oversight, meaning there is no opportunity for improvement as an author.

**Support:**

3

---

> ### Author Rebuttal · Authors · 2026-03-28
>
> We thank the reviewer for the good words. We are particularly encouraged by the point you raised: *"it can spark a lot of discussion that will benefit the community."* Frankly, no one could design a perfect system in one shot, and us included, but we do believe our proposed credit system is flexible enough to evolve, correct, and benefit the community.
>
> Since you did not raise any concern, we took the liberty of replying to `AJtz` here, where many interesting discussions were raised. We apologize for this slightly awkward workaround due to character limits.
>
> ---
> ## **`Reviewer AJtz's W1, W3.2, W4, Q1, Q2 - Different threat models and attack vectors` Interesting scenarios; here are our proposed defenses.**
>
> We will be the first to admit that our system can be gamed, attacked, or shortcut. Even in the real world there is no shortage of financial crime; so we do not pretend our 8-page position paper is ironclad, and we say so explicitly in `L077L`, `L266R`, `L610`, `L636`, and more. Our argument is that, **although imperfect, a credit system has a much better fighting chance to these attack attempts, because organizers can award and deduct points to impose countermeasures and incentivize good behavior.**
>
> Due to character limits, we cannot address every scenario you raised. But at a high level, the goal is to make attacks sufficiently difficult through *feasible safeguards*, while *aligning incentives* so that the right behavior is also the easier one. In `App. B: FAQ - How do we prevent bad behavior (fraud, point farming, unfairness, ...)? We already introduced several countermeasures. Here, we group them into a four-part defense:
>
> 1. **Duty Delegation** — restrict certain actions to specific roles.
> 2. **Upper Limits** — cap how often an action can be taken.
> 3. **Dynamic Pricing** — increase cost with repeated usage.
> 4. **Voting-based Penalties/Awards** — enable peer-driven judgments.
>
>
> Below, we discuss several scenarios you raised:
>
> * `W1.1` *Bulk outsourcing*
>
>   * **#2** limits how many papers one can review per cycle.
>   * Under **#3**, services such as repeated "additional expert" requests can become progressively more expensive.
>   * Under **#4**, low-quality outsourced reviews may be penalized, making outsourcing less attractive than writing good reviews and aiming for rewards.
>   * Above designs can also help mitigate the concern of `W3.2`.
>
> * `W1.2 & Q1` *Large entities appointing a dedicated "point person" as leading author across many papers*
>
>   * Leveraging **#1**, submission fee-like costs can be restricted to leading authors (e.g., first or last authors). We doubt real lead authors would often give up these positions to a dedicated "point person," when they can just properly review a few papers.
>   * Under **#3**, we can make submissions > X papers progressively more expensive.
>   * Per PaperCopilot statistics for ICLR 2025, the top individual author coauthored 42 papers, while the top institution was associated with 265 submissions. Without P2P transfer, it would be difficult for one person to accumulate enough points at that scale, and the optics would also be poor (people get called out for coauthoring way less).
>
> * `W1.3 & 1.5` *The "additional expert reviewer" option being spammed.*
>
>   * **#2**: a hard cap can be set on how many requests an another can make.
>   * **#3**: later requests could become increasingly expensive, encouraging authors to reserve them for their most important submissions.
>   * We also note that non-monetary perks matter (see `L553`), since otherwise well-resourced actors may remain insufficiently motivated.
>
> * `W1.4 & Q2` *Large entities lobbying for money-to-points conversion / P2P transfer*
>   * We want to state clearly that rejecting P2P transfer and  point buying is crucial. Much of our proposal relies on the principle of "**(good) labor for perks**," which is directly undermined if points can simply be bought.
>   * For this same reason, if point buying were successfully lobbied into the system, we would expect the community to seek alternative venues, because it would have effectively become pay-to-play, which is probably worse that what we have now.
>
> * `W4` *Doing only the bare minimum to earn enough points for the next submission cycle*
>
>   * We do not deny this possibility; there will always be "slackers." However, the dynamic nature of **#4**, along with meaningful rewards for outstanding reviewing, creates pressure to exceed the minimum.
>   * Further, by allowing reviewers to "buy out" or preemptively fulfill reviewer obligations in cycles when they are not core submitting authors (Sec. 5.3), we give them a better workload-management option when they are doing their own rebuttals.
>
> Overall, we readily acknowledge that no system can be made perfect in one shot. But we hope **the discussion above shows that a credit system is flexible enough to offer concrete tools against many concerns, whereas current conference mechanisms often lack such levers altogether.**

---

> > ### Author Rebuttal · Reviewer_Fd3q · 2026-04-02
> >
> > Thank you for pointing out that your system can be improved and corrected. I will mantain my score.

---

### Official Review · Reviewer_AJtz · 2026-03-10

**Significance:** 3
**Argument Clarity:** 2
**Rating:** 4
**Confidence:** 3

**Questions:**

1. A points-based system might provide additional attack surface for the large sponsors and companies to influence the reviewing process. How can we mitigate these risks?
1. A strict regulation of points transfer might be considered undemocratic by the community (similarly to the strict regulation of wealth transfer in real life). What is your stance on this matter?

**Alternative Views Section:**

Yes

**Compliance With Llm Reviewing Policy A Conservative:**

Affirmed.

**Discussion Potential:**

3

**Final Justification:**

Most of my concerns were addressed during the rebuttal, but the scale of the changes is quite substantial, and their actual implementation in the text would benefit from another round of review.

**Paper Summary:**

To address the many problems associated with peer review, the authors propose adopting a point-based system in which points can be earned through reviewing or otherwise contributing to the community, and then spent on submissions, exemptions, additional reviews and other useful perks.

**Position:**

Yes

**Position In Title:**

Yes

**Related Work:**

2

**Strengths And Weaknesses:**

**Strengths:**

1. The problem considered in this paper is very important and relevant to the ICML (or any) research community.
1. The paper is well-organized.
1. The "Alternative Views" section already covers many of the possible questions.
1. Practical nuances are addressed in Section 6.

**Weaknesses:**

1. Despite the authors proposing a market-based solution, important market-related problems remain unaddressed (I am aware of the "How do we prevent bad behavior" paragraph in the Appendix, which I find very brief).
   - **"Black market"**. In Section 5.8, the authors advocate against using real money for their system. However, because rewards and punishments are formalized in terms of points, it is logical to expect that money will find a way to be converted into them. I strongly suggest the authors research such ways preemptively. Below, I provide a non‑exhaustive list of possible attack vectors (note that they are not necessarily the most feasible or well‑thought‑out):
      - A point-based reward system creates additional incentives for reviewers to pay a third party to perform the review.
      - Large companies or research groups may massively outsource reviewing, with formal points receivers being the leading authours who appear on every paper. These authors therefore can accumulate large amounts of points and still "spam" low-effort papers until acceptance. Moreover, the "additional expert" option (or similar) can also be frequently utilized to steer the outcome of papers that receive low scores.
      - Large entities and conference sponsors may lobby for introduction of a direct money-to-points conversion. Furthermore, the conversion rates may ultimately become unaffordable for non-corporate entities.
   - **Rich actors favor paying with money**. Perks that otherwise would require real money (e.g., registration) induce a systematic disadvantage for independent researchers: due to objective circumstances, they are more likely to spend their points on money-related perks, rather than on exemptions, additional reviews or other non-money-based perks.

   Overall, a significant effort should be made to facilitate a thorough discussion on **whether the proposed system favors large entities over small research groups**.
1. The supporting evidence is often anecdotal, referring the reader to sporadic sources (e.g., a "Who is Adam?" tweet, an occasional LLM-written paper that got accepted to a top-tier conference, or the OpenReview security incident). This source material is unsystematic and cannot be used directly for a scientifically rigorous discussion.
   - **The OpenReview security incident** is hardly a good illustration of the claim "almost everyone has many unpleasant things to share about their review experience" (lines 019-020), since the scale of abuse was not quantified in any meaningful way: apart from the 45% figure for the fraction of leaked metadata provided in the blog post, no concrete data on the incident is publicly available. Therefore, there is no quantitative basis to describe the "tension between authors and reviewers".

     Perhaps, an extensive survey, poll, or sentiment-analysis study should be used to better substantiate the authors' position.
   - To illustrate the **"AI-assisted research becoming more accessible"** (lines 089-090), the authors refer to a single ACL 2025 paper by Zochi, Intology’s Artificial Scientist.
      - This AI agent is deployed by a corporate entity and, therefore, does not directly illustrate the accessibility of AI-assisted research (since an average researcher likely does not possess the resources and tools comparable to those of a company that focuses on AI-generated papers).
      - More importantly, some statistics and long-term observations should be reported or cited to convince the reader that this case is not just a coincidence arising from the growing number of submissions. For example, a (possibly) growing fraction of significantly AI-assisted accepted papers should better illustrate the author's point, given they are able to collect the supporting data.
   - **Some important points are not substantiated at all with any external evidence**. For example, the claim "submission caps mostly change who gets listed on a paper, rather than whether the paper is submitted" from Section 3.1 or the claims from Section 3.3 "100% Mandatory Reciprocal Reviewer Recruit is a Slow-Acting Poison".
1. The article is clearly inspired by the scalability-related problems: the appeal to ever-growing number of submissions and increasing reviewers', ACs', and SACs' workload appears many times throughout the paper (e.g., Section 2.1). However, the discussion of how well the proposed system scales with the number of submissions is very limited to say the least. The authors suggest a points-based mechanism for controlling the number of submissions, but do not discuss whether the whole proposed system degrades gracefully when the number of submissions still keeps increasing. The following discussion directions are therefore strongly suggested:
   - How hard is it to moderate a point-based system with a growing number of submissions?
   - Can a points-based system still be abused by AI tools (e.g., via doing cheap and low-effort tasks in bulk with LLMs)?
   - Can poor scaling lead to other market-related problems (like inflation of points)?
1. The authors provide a decent critique of existing approaches, but sometimes fail to apply the same arguments to their proposal. E.g., on lines 183-188, one can find the following:

   > if a reviewer/AC/SAC is not self-motivated to perform well but is instead forced to fulfill a duty, such a person can always find a way to do the bare minimum, regardless of how meticulous the policy is.

   However, the possibility of a reviewer/AC/SAC doing the bare minimum to acquire just enough points for the next publishing cycle is not discussed. I find this quite possible under large workloads: rather than risking their time and effort for additional points, a reviewer might prefer doing the bare minimum and spend more time on their own rebuttal.
1. Similar proposals are also being discussed in the open source community (e.g., the `vouch` system, with different mechanisms for controlling the vouching list, including point-based). These approaches should be added to the overview and studied.

**Typos, minor issues and errors:**

1. Footnote 11 is split into two parts, one on page 4 and the other on page 5.

**Rating motivation:**

While the idea is interesting and will most certainly spark a discussion, the presented arguments and alternative views should be improved before the publication.

**Support:**

2

---

> ### Author Rebuttal · Authors · 2026-03-27
>
> We find many of your raised threat models interesting, and we appreciate that you spent time thinking about those. Due to space limitations, we kindly redirect you to [our reply to Reviewer `Fd3q`](https://openreview.net/forum?id=WwrlJMwTJL&noteId=Qtrw6dRsre) for discussion regarding such attack scenarios.
>
> ---
> ## **`W2 - Anecdotal evidence is weak.` Agreed, here are some alternative sources.**
> * *Unpleasant experiences in ML review.*
>   * To provide more quantitative and fewer cherry-picked supports, we will additionally cite the NeurIPS 2025 D&B Author Exp Survey, J&R (2022) *What Factors Should Paper-Reviewer Assignments Rely On?*, and the NeurIPS consistency exps. Statistics such as roughly 25% of surveyed authors explicitly reporting poor review quality, and roughly 50% of decisions flipping if papers were re-reviewed, shall provide more quantitative support.
> * *AI-assisted research becoming more accessible.*
>   * We agree that the Zochi paper is more of a 0-to-1 of AI-scientists making main conference and, by itself, provides not much support about accessibility. Broader accessibility support should instead come from Yang et al., Kim et al., and Eger et al. (2025), which contain more explicit signals, such as the increasing prevalence of AI-related wording and usage patterns. We already cite these works around the same paragraph, but we should make their findings more explicit in our writing.
>   * AI-hallucinated citations, recent `autoresearch` trends, Opus 4.6 1M moving from Extra Usage to Subscription,  Lu et al. *Towards end-to-end automation of AI research* published this week at Nature, may also help support the growing adoption of AI assistants in research.
> * *On whether "submission caps mostly change who gets listed on a paper."*
>   * Our intuition is that very few authors are core contributors (e.g., first/last authors) on a large number of submissions, whereas many authors do appear on a large number of papers overall. E.g., PaperCopilot ICLR25 statistics show:
>
> |           Criteria           | Top #1 | #25 | #50 | #75 | #100 |
> | :--------------------------: | :----: | :-----: | :-----: | :-----: | :------: |
> | Most submitted First Authors |    7   |    4    |    3    |    3    |     3    |
> |         ~Last Authors        |   34   |    13   |    11   |    9    |     8    |
> |         ~Any Authors         | **42** |  **21** |  **17** |  **16** |  **14**  |
>
> The gap between the third row and the first two is substantial. While we can never know whether the contributions of those "any-author" cases were core enough that the paper could not reasonably be submitted without them, we believe it is still reasonable to infer that it is unlikely many authors are deeply involved in that many papers per cycle. If a harsh cap were imposed, some authors might therefore choose to remove their names from lower-priority coauthored papers in order to prioritize others. That said, you are right that we do not have direct evidence for this claim, and we will clarify that more explicitly in the revised version alongside the table.
> * *On mandatory reciprocity.*
>   * Since this concerns reviewers' mental framing, there is likely no direct evidence specific to this exact mechanism. We shall concede here and use much softer language like *"when recruited by force and with limited motivation and bandwidth, reviewers are likely less willing to go the extra mile (e.g., initiating internal discussions), as their main drive is to fulfill their assigned duties so their own submitted works won't get desk rejected."*
> ---
> ## **`W3 - Scaling of the credit system.`**
>
> From our view, there are two ways to address this:
> 1. We model and simulate our proposed system, and show that it would scale.
> 2. We show that some integrable designs would help with scaling challenges.
>
> We don't think #1 is possible with currently available technology, for reasons discussed in Sec 7 and our response to `Gc4s`' `Q1`. However, for #2, we do think there are ample tools available that could help with such concerns, such as:
>
> * The credit card point system provides many established designs for point moderation and inflation control — e.g., by providing discounted service vouchers under limited-time windows and setting expiration dates, banks encourage people to spend their points — which can be borrowed.
> * Similarly, many malicious behaviors can be deterred and mitigated by the four-point defense we shared under `Fd3q`.
>
> ---
> ## **`W5 - mitchellh/vouch as related work`**
>
> No space left, but thank you for raising this. Vouch’s “web of trust” and its vouch/denounce features are indeed conceptually related to our credit system. The key difference might be `vouch` is much more front-door-focused with an access-control-oriented design, whereas our proposed system is aimed more at shaping behavior after admission (for obvious fair science reasons), with greater flexibility in how earned points can be spent. We shall discuss more in the updated manuscript.

---

> > ### Author Rebuttal · Reviewer_AJtz · 2026-04-03
> >
> > I thank the authors for their thorough responses. I particularly appreciate the specificity of the proposed amendments, as the original manuscript gives the impression that the authors were very hesitant to be specific (e.g., with phrases like "to be clear, we are not arguing for the enforcement of any specific rule"). I strongly encourage the authors to incorporate all of this rebuttal material into the final revision of the paper.
> >
> > I still believe that additional research is needed to fully address W3, but I am satisfied with how the other weaknesses have been handled. Consequently, I am raising my score to "Borderline Accept". I am not assigning "Accept" because the scale of the changes is quite substantial, and their actual implementation in the text would benefit from another round of review.
> >
> > If the authors still have some time, I would also appreciate an answer to my Q2.

---

### Official Review · Reviewer_Kc3H · 2026-03-10

**Significance:** 4
**Argument Clarity:** 3
**Rating:** 4
**Confidence:** 4

**Questions:**

see comments

**Alternative Views Section:**

Yes

**Compliance With Llm Reviewing Policy A Conservative:**

Affirmed.

**Discussion Potential:**

4

**Final Justification:**

I support the paper and also agree with the proposed position. I do believe it would have been better if the authors had a demo of something that works (even in toy settings). In all, I support the paper for acceptance.

**Paper Summary:**

Argues for an update in the paper review system. Authors posit that polite requests and guidelines don't work, and we need stricter, fine-grained and enforceable procedural guardrails. They advocate using a spendable, across-conference credit system for a sustainable review ecosystem with right incentives.

**Position:**

Yes

**Position In Title:**

Yes

**Related Work:**

3

**Strengths And Weaknesses:**

Strengths:
- Extremely relevant issue given the dire state of ML conference peer-review system.
- Highlights valid root causes of poor reviews, and has reasonable takes on those why current ways to tackle them are not very helpful (would consider renaming section 3 title).
- Detailed discussion of alternative views and reasonable calls for action.

Weaknesses/Comments:
- I have seen this position floated around multiple times, in some form or another, at conferences or otherwise (online discussions). To make it a good position paper and be discussed seriously at ICML, I expect a proper exact incentivizing mechanism thought out in detail. And an alternative views section discussing some potential pitfalls of the proposed mechanism and how to mitigate them (this could potentially be improved during the review of this position paper). Writing a footnote (11) saying that we do not propose exact details but just call for action, is not very meaningful (I've seen multiple position paper track papers, which only live to become a poster at the conference and see no meaningful movement after that). For this reason I don't like section 4.1, it should be properly laid out with a functioning mechanism. In contrast, I like section 4.2, which proposes an actionable item, discusses potential problems (which should have gone into alternative views) and still argues for why that is acceptable. The paper is lacking a stronger section 4 (the proposals).
- FAQs from the appendix should be included in the development of a strong section 4.
- 3.1: I agree with "... endlessly recycling previously rejected work with critical flaws" point, but don't know if there is evidence of "submission caps mostly change who gets listed on a paper". If I worked on a paper, I would not be OK in getting removed from the author list, or getting named on another paper where I didn't contribute. Its hard to work out how many people are unethical like this.
- 3.2: "From a reviewer’s perspective, the only thing that matters is their own current or future submissions." -- I don't think so. This can be true for an increasingly large number of reviewers nowadays, but it is not how every reviewer behaves.
- [Minor] I don't remember the last time I read so many em-dashes in an article. Excessive and many a times unnecessary (e.g. L132, L139, L257-8, etc.). Gives an AI "smell" to the writing (I did not run the content through a AI generation check, and I believe the content is mostly written/supervised by human(s)). Recommend looking into when to use em-dashes and when to use simple commas.

I am recommending a weak accept to show that I care about this issue and would like to see a stronger section 4 with exact mechanisms to recommend proper acceptance.

PS: Who is Adam though? :)

**Support:**

2

---

> ### Author Rebuttal · Authors · 2026-03-25
>
> This is a non-technical work, so we intend to make this less of a rebuttal and more of a discussion, as we believe we have all experienced some unpleasantness in ML review. Rather than arguing that our way is better, we are more interested in learning your concerns and how we can mitigate them in our proposed system.
>
>
> ---
>
> ## **`W1&2 - Sec 4: More exact incentivizing mechanism details.` Sure, here is a plan for improvement.**
>
> Our take on the proposed credit system is:
>
> >`L82` We are also not there to propose a specific set of rulebook that all conferences must follow, but rather to advocate for a promising general direction that future conference organizers can explore and adapt to their own needs.
>
> This prevents us from being too exact, since:
>
> * We don't want a my-way-or-highway proposal, but a flexible design that adapts to different conferences' needs.
>     * After all, one major point of having currency is the freedom it offers — if we don't like one store, we'll buy from another one— and we intend to keep it that way for "conference shopping".
> * There are things we lack data to be exact on, e.g., in Sec 6.3, we discussed why we can't determine exact point-values for contributions and perks, as critical review statistics are not cleanly available.
>
> **That said, we strive for greater exactness where we can.** E.g., we discuss many actionable recipes with quite a bit of exactness involved in Sec 6. It seems you are **mainly dissatisfied with Sec 4.1, as it lacks actionable items.** If so, would you like this improvement plan?
> * List out and expand on several proposed implementations in Sec 4.1.
>     * Authors writing reviewer feedback for points.
>     * Refundable "submission fee" to reduce submission load.
>     * Opt out of otherwise assigned review duties (from Sec 5.3).
>     * Mobilizing researchers not submitting as main authors to participate more (from Sec 5.3).
> * Include more operational details from App B FAQ.
> * Elaborate on what we can do to make it more friendly to less-supported scholars, such as Sec 5.6 and our discussion with `Gc4s` and `AJtz` below. With major ones in Sec 4 and more specific ones in Sec 5.3.
>
> We can leverage the extra camera-ready page for these additions, and optionally push Sec 7 to the app.
>
> ---
>
>
> ## **`W3&W4 - Overly broad claim regarding submission cap and reviewer mental framework.` We agree that those points are not well supported and will soften the language.**
>
> We agree these claims, while minor, are a bit carelessly made. There is no way we can know the reviewer's mental activity, so it would be pretentious to defend such takes. More appropriate language here would be:
>
> > Should force-recruited reviewers want to take an ultra-pragmatic view and treat the review duty as a mere to-be-finished task, the things that matter most to them are likely their own current or future submissions.
>
> On whether *"submission caps mostly change who gets listed on a paper"*: our train of thought is that there is very little chance an author is the core contributor (first/last authors) of many submissions, but many authors do have their name on a lot of papers. This is evidenced by PaperCopilot, let's take ICLR25 statistics for example:
>
> | Criteria |  Top #1 | Top #25 | Top #50 | Top #75 | Top #100 |
> |:-:|:-:|:-:|:-:|:-:|:-:|
> | Most submitted First Authors | 7 | 4 | 3 | 3 | 3 |
> | ~Last Authors | 34 | 13 | 11 | 9 | 8 |
> | ~Any Authors | **42** | **21** | **17** | **16** | **14** |
>
> The gap between the third set and the first two is significant. While we can never know if these third-set authors' contributions are "core-enough" to prevent a paper's submission without their name, we believe it is reasonable to deduce that it is unlikely for many authors to have core involvement with that many per cycle. Thus, they might be willing to strip their name from some coauthored papers to prioritize others, if a harsh cap is set. But you are right that there is no explicit evidence, and we will clarify this in the updated version alongside this table.
>
> ---
>
> Last, **making this design heard, discussed, and maybe implemented is the sole goal of this paper.** A non-technical paper's acceptance provides little influence on our career, but a better conference review mechanism helps all of us. We are invested in not making this a forgettable poster; in fact, we have even raised this idea (in a more preliminary form) to some ex-SPC/PC of ICML/NeurIPS/ICLR, who replied like:
>
> > Thank you so much. A currency or credit-like system is indeed something that I have heard from other community members. I fully agree that a more positive cycle of reciprocal participation is needed. I'll record this idea for future program chairs to think about. Many thanks.
>
> ---
>
> (And we will fix up regarding em dash. As for Adam, we heard this guy has something to do with a lady with a nightly name, a fruity tech company that makes the iPhone, and a linear sequence model called Mamba. Not sure if we are hallucinating though :)))

---

> > ### Author Rebuttal · Reviewer_Kc3H · 2026-04-02
> >
> > OK I can support this paper for acceptance. If you make it to the conference, your poster/presentation should strive to highlight your actionable items to incite discussion. Good luck!

---

### Decision · Program_Chairs · 2026-04-30

**Decision:**

Accept (regular)

**Comment:**

Reviewers need to be incentivized.  Yes.  This is well known.  So what of the authors' discussion and proposal.

The authors discuss what they believe are the causes and also the flaws of some existing attempts to incentivize.  Reviewers are mixed on these.
The authors propose a points based system.  This would seem to be fraught with potential problems, but, well, maybe its worth a test.  This is definitely a position for discussion, and the reviewers provide some examples of that.
But the writing and arguments needed tightening.
So this is borderline, but such a substantial problem, with a candidate solution, it makes a good position.